



# Subseasonal Forecast Improvements from Sea Ice Concentration Data Assimilation in the Antarctic

Yong-Fei Zhang[1], Mitchell Bushuk[2], Michael Winton[2], William Gregory[1], Bill Hurlin[2], Liwei Jia[2,3], and Feiyu Lu[1]

[1]Atmospheric and Oceanic Sciences Program, Princeton University, Princeton, New Jersey
[2]National Oceanic and Atmospheric Administration/Geophysical Fluid Dynamics Laboratory, Princeton, New Jersey
[3]University Corporation for Atmospheric Research, Boulder, Colorado

**Correspondence:** Yong-Fei Zhang (yongfeiz@princeton.edu)

**Abstract.** This study evaluates the impact of sea ice concentration (SIC) data assimilation (DA) on subseasonal forecasts of Antarctic sea ice by comparing reforecast experiment suites initialized from two sets of initial conditions (ICs): one with SIC DA and the other without. The two ICs are evaluated against NSIDC SIC observations. Results show that the SIC DA

significantly improves the climatology and interannual variability of the SIC IC. The improvement in sea ice ICs is more considerable in the Antarctic than in the Arctic. The sea ice thickness (SIT) field is mostly thinner after SIC DA except for the interior Weddell and Ross sectors. The results from reforecast experiments show that SIC DA improves the subseasonal forecast skill of Antarctic SIC in almost all initialization months except December and January, where the initial improvement is soon overtaken by the bias likely linked to the thin SIT bias. We also demonstrate that SIC DA improves the probabilistic

prediction of the sea ice edge position at subseasonal time scales. The subseasonal reforecast skill of Antarctic SIC and the sea ice edge is improved the most in spring, followed by winter and summer, and has minor differences in autumn. The skill improvement associated with SIC DA is more significant in the Antarctic than the Arctic, consistent with the IC improvement. Our study demonstrates the critical role of SIC DA in the subseasonal prediction of Antarctic sea ice.

## 1  Introduction

The Antarctic sea ice extent (SIE) experienced a steady increase from the late 1970s to 2015 (e.g., Zwally et al., 2002; Cavalieri and Parkinson, 2008; Parkinson and Cavalieri, 2012; Turner et al., 2015; Zhang et al., 2019), as opposed to the continuously declining Arctic SIE. In the austral spring of 2016, the Antarctic sea ice started retreating at an unprecedented rate that led to the lowest summer SIE record in 2017 in the satellite era (e.g., Stuecker et al., 2017; Turner et al., 2017; Schlosser et al., 2018; Zhang et al., 2022a). The Antarctic SIE remained abnormally low until the present and reached record minimum lows in

February 2022 (e.g., Raphael and Handcock, 2022), and then again in February 2023 (e.g., Purich and Doddridge, 2023). While Antarctic sea ice had not received as much attention as its northern-hemisphere counterpart, its shift from the steadily increasing trend to the recent decline stimulated a growing research community (e.g., Massonnet et al., 2023). In the meantime, human

---

[1]The corresponding author's present address: Earth System Science Interdisciplinary Center, University of Maryland, College Park, Maryland



activities, including commercial shipping, fishing, and tourism, have been accelerating in the past decades. Hence, providing more accurate subseasonal predictions of Antarctic sea ice has become pressing to better manage fast-growing human activities.

Previous studies have proven the value of initialization in Antarctic sea ice predictability at seasonal to decadal time scales. Holland et al. (2013) show, in a perfect model study, that the initial value of ocean and ice states in January provides predictability of the sea ice edge for up to 3 months and then re-emerges in the ice growth season due to the memory of heat content anomalies stored in the upper ocean. Zunz et al. (2015) explore the impact of three different data assimilation approaches to constraining sea ice and ocean initial conditions on Antarctic sea ice predictions at interannual to decadal time scales within

a perfect model framework initialized from January. They find the benefit of the initialization shown in the high prognostic potential predictability and correlation with observations in the first two months, which disappears and re-emerges in the later season, attributed to the memory stored in the upper ocean heat content. They also demonstrate that the benefit of initialization may even exist in decadal time scales likely due to the interaction between sea ice and ocean heat content. Marchi et al. (2019) further employs six GCMs and conducts ensemble experiments on multiple start dates to study the initial-value predictability

of Antarctic sea ice. They confirm that even though they initialized the models from a different month, July, this initial skill drop in the first couple of months and skill re-emergence linked to the upper ocean heat anomalies in the subsequent winter season still exists.

   The mechanisms of Antarctic sea ice predictability at seasonal to interannual time scales are similar to that of the Arctic in that the anomaly persistence of sea ice itself explains the majority of the predictability source at the lead times up to a couple

of months, and the upper ocean heat anomalies contribute to predictability at the longer lead times, as shown in the above studies. While studies agree that sea ice thickness (SIT) is an important predictor for Arctic summer sea ice predictions (e.g., Blanchard-Wrigglesworth et al., 2011; Tietsche et al., 2014; Guemas et al., 2016; Bushuk et al., 2021), it is less conclusive for the role of SIT in Antarctic summer sea ice predictions at seasonal time scales. While some studies (e.g., Holland et al., 2013; Ordoñez et al., 2018; Marchi et al., 2019) suggest that SIT is not an important predictor for Antarctic summer sea ice

predictions, Bushuk et al. (2021) found that SIT is a strong source for summer sea ice predictions in the Weddell Sea.

   Whether prediction systems can turn the potential predictability into actual prediction skill is even less studied. Several hindcast studies have been done to assess it in the current seasonal prediction systems. Bushuk et al. (2021) show that the GFDL seasonal prediction systems can make skillful predictions of regional SIE and sea ice edge positions at seasonal time scales. They also demonstrate that the new systems (SPEAR_LO and SPEAR_MED) have improved summer sea ice predictions

over the previous system (FLOR), mainly due to the more accurate initial conditions of SIE and SIT. Payne et al. (2023) found similar results by conducting analyses on the seasonal prediction skill of Antarctic sea ice in the Canadian Seasonal to Interannual Prediction System version 2 (CanSIPSv2). They also compared the seasonal prediction skills between the two hemispheres and found that the skill difference varies with the experiment period. The detrended skill is generally better in the Antarctic than in the Arctic from 1980 to 2019, while it's the other way around when analyses are done from 1980 to 2010.

Studies have also examined the Antarctic sea ice prediction skill at subseasonal time scales. Zampieri et al. (2019) evaluated subseasonal forecasts of the sea ice edge position from the ensemble prediction systems in the S2S database from 1999 to 2010 in both hemispheres. Only one system outperforms the benchmarks (anomaly persistence and climatology forecasts) in the



5–30 day forecast range. All systems perform worse in the Antarctic than in the Arctic, likely linked to the substantial initial errors in the Antarctic. Gao et al. (2024) extended the study of Zampieri et al. (2019) in the Antarctic by updating to the most recent S2S database and including prediction systems from the Copernicus Climate Change Service (C2S) project. Similar results are shown in their study regarding the performance of the systems in Antarctic sea ice edge predictions. They further diagnosed possible causes of the sea ice edge errors in the systems and highlighted the critical role of initial conditions (ICs). Wang et al. (2023) constructed a deep learning model for Antarctic regional sea ice at sub-seasonal time scales using only daily SIC from observations and showed that it outperformed the three major seasonal prediction systems (GFDL, ECMWF, and NCEP) that were examined in their study, highlighting errors in the initial condition of the dynamical models.

We have previously demonstrated the value of SIC DA in improving sea ice ICs in the Arctic using the sea ice DA framework of the GFDL ocean and sea ice model and the Data Assimilation Research Testbed (DART) (Zhang et al., 2021). We further examined its value in seasonal Arctic summer sea ice predictions using the GFDL seasonal prediction system (SPEAR) (Zhang et al., 2022b). Gregory et al. (2023, 2024) also show that SIC DA reduces systematic IC biases of Antarctic sea ice in the GFDL sea ice model. Inspired by the growing body of studies on Antarctic sea ice prediction and predictability, in this study, we thoroughly evaluate the performance of an updated version of the SIC DA product generated from our sea ice DA framework in the Antarctic. We then investigate the benefits of using the SIC DA product to initialize the GFDL SPEAR system for Antarctic sea ice prediction at subseasonal timescales. We describe the data and methodology in Section 2, analyze and discuss results in Section 3, and conclude in Section 4.

## 2 Data and Methodology

### 2.1 SPEAR System

The current GFDL modeling system for seasonal to multi-decadal predictions and projections, the Seamless System for Prediction and EArth System Research (SPEAR) (Delworth et al., 2020), is applied in this study. It consists of the Modular Ocean Model version 6 (MOM6), Sea Ice Simulator version 2 (SIS2), Atmosphere Model version 4 (AM4), and Land Model version 4 (LM4). SPEAR has two initialized seasonal prediction systems: SPEAR_LO and SPEAR_MED. They differ in their horizontal resolution in the atmosphere and land models, with SPEAR_LO having $1°$ and SPEAR_MED having $0.5°$. They share the same nominal horizontal resolution of $1°$ for ocean and sea ice models. SPEAR_LO and SPEAR_MED are found to have similar seasonal prediction skill for Arctic sea ice (Bushuk et al., 2022), but the latter was shown to perform better for Antarctic sea ice (Bushuk et al., 2021). Thus, we employ SPEAR_MED in this study.

Two separate data assimilation experiments are conducted to provide initial conditions (ICs) for the SPEAR prediction system. An ocean DA experiment performed using SPEAR_LO provides this ocean ICs. This experiment assimilates observations of sea surface temperature (SST) and ocean temperature and salinity profiles from various sources (Lu et al., 2020). An atmosphere and SST nudging run based on SPEAR_MED provides the sea ice, atmosphere, and land ICs. This experiment nudges the wind, temperature, and humidity to the NOAA/NCEP Climate Forecast System Reanalysis and SST to NOAA's Optimal Interpolation Sea Surface Temperature (OISST). The SST is nudged to OISST when the observed SIC (also from OISST) is



below 30%, and to the freezing points calculated based on the modeled salinity when the observed SIC is >30%. Hence, the sea ice IC is implicitly constrained by SIC observations.

## 2.2 SIC DA Product

We assimilate passive microwave SIC observations retrieved using the NASA Team (NT) algorithm from the National Snow
and Ice Data Center (NSIDC) version 1.1 into the SPEAR MOM6/SIS2 global ice-ocean model forced by the Japanese 55-year Reanaysis (JRA55-do) (Tsujino et al., 2018), through the linked Data Assimilation Research Testbed (DART)(Anderson et al., 2009) and SIS2. An Ensemble Adjustment Kalman Filter (EAFK)(Anderson, 2001) with a horizontal localization radius of approximately 170km was used in this study. The technical details of the assimilation system are documented in (Zhang et al., 2021). This SIC data assimilation product has been validated in the Arctic (Zhang et al., 2021) and shown to improve
subseasonal-to-seasonal predictions of Arctic summer sea ice in a subsequent study (Zhang et al., 2022b).

## 2.3 Reforecast Experiments

In this section, we describe the two seasonal reforecast experiment suites analyzed in this study. Since the SPEAR_MED system has been shown to have better prediction skill for Antarctic sea ice Bushuk et al. (2021), we employ the SPEAR_MED instead of SPEAR_LO. Our baseline reforecast suite is the standard SPEAR_MED forecast described in Section 2.1. Hereafter,
we refer to SPEAR_MED as SPEAR for simplicity. The other reforecast experiment, SICDA, is initialized from the SIC DA product described in 2.2. SICDA has the same initial conditions for the ocean, atmosphere, and land as SPEAR. We refer to their ICs as SPEAR_IC and SICDA_IC, respectively.

## 2.4 Reference Forecasts

We use the damped anomaly persistence forecast (damped persistence hereafter) as the baseline reference forecast. The damped
persistence forecast is calculated using the NSIDC NT SIC observations for each grid cell and consists of two components: a linear trend climatology and a scaled anomaly. For a given grid cell on a given target date, we calculate the slope and intercept using the observed SIC on this day in all the past years (e.g., 1979 to the year before the target year) to compute the linear trend climatology. The scaled anomaly is computed by multiplying the observed anomaly on the day before the initialization date by a scaling factor. The scaling factor considers the lagged correlation between the observed SIC on the initialization date and the
target date. The detailed equation can be found in Equation (5) in Bushuk et al. (2024).

## 2.5 Observation and Reference Data

The NSIDC SIC observations derived using the NT algorithm are used in this study. The NSIDC NT SIC observations are assimilated into the SICDA_IC and are also used to evaluate the results of the reforecast experiments.

We also consider sea ice thickness (SIT) analysis data from the Global Ice-Ocean Modeling and Assimilation System
(GIOMAS; Zhang and Rothrock,2003) as a reference data set. The GIOMAS SIT data is not an observation data set, nor





has it assimilated any SIT observation data. Previous studies (e.g., Liao et al., 2021; Shi et al., 2020) evaluated it against satellite retrievals, shipping and airborne observations and concluded that the GIOMAS SIT field can reasonably represent the climatology, interannual variability, and trends of Antarctic sea ice, although it does suffer from biases. It was found to underestimate SIT, especially in the deformed ice zone, for example, the northwestern Weddell Sea. Note that the GIOMAS

SIT data set is not used for model validation in this study, but rather a reference, as has been used in other Antarctic sea ice studies (e.g., Shu et al., 2015; DuVivier et al., 2020; Selivanova et al., 2024).

Another SIT reference dataset we use is from the Southern Ocean State Estimate (SOSE) that comes from the Southern Ocean Carbon and Climate Observations and Modeling (SOCCOM) project (Mazloff et al., 2010). The most recent endeavor includes a biogeochemical component (BSOSE), which assimilates all the available physical and biogeochemical ocean ob-

servations from floats, ships, and satellites in the Southern Ocean into the MIT general circulation model (Verdy and Mazloff, 2017). Although BSOSE does not assimilate any SIT observations, its constraints from ocean profiles, SST, and SIC observations should provide a good candidate for reference. The monthly SIT data from the 105th iteration of the BSOSE dataset is used in our study. Note that both datasets provide the ice thickness per grid cell, which is referred to as sea ice volume (SIV) in convention.

## 2.6   Skill Metrics

We evaluate the regional SIE and grid-cell level SIC from the two ICs. The regional SIE is the sum of the area where SIC is greater than 15%. The Antarctic ocean is divided into five sectors following the convention in Bushuk et al. (2021): Weddell (60°W–20°E), Indian (20°–90°E), west Pacific (90°–160°E), Ross (160°E–130°W), and Amundsen and Bellingshausen (130°–60°W; A&B hereafter). We evaluate the two model ICs using the detrended anomaly correlation coefficient (ACC), bias,

and Root-Mean-Square Error (RMSE). We first take the ensemble mean of SIC from the 30 members of each initial condition data set. We then calculate the skill metrics using the ensemble mean. We use the detrended ACC and RMSE to evaluate the reforecast skill of SPEAR and SICDA as well. The detailed equations are documented in the Section 2d in Zhang et al. (2022b). When we look at the skill of grid-cell level SIC averaged for sub-regions, we only average skills within the ice variability zone. We define the ice variability zone as the grid cells where the observed interannual variability is larger than 10%.

In addition, we calculate the Spatial Probability Score (SPS; Goessling and Jung (2018)) to evaluate the sea ice edge predictions of SPEAR and SICDA. We define SPS as

$$SPS = \int_s (P_f(i) - P_o(i))^2 \, ds$$

where $P_f(i)$ and $P_o(i)$ are the sea ice probabilities at the grid cell $i$. $P_f$ is the percentage of ensemble members with larger than 15% SIC for the SPEAR ensemble forecast and either 0 (SIC<15%) or 1 (SIC>15%) for the deterministic damped per-

sistence forecast. $P_o$ is also 0 or 1 since observations are deterministic as well. S is the integral surface area evaluated over a spatial domain (Arctic or Antarctic ocean). Since the Arctic and Antarctic have different ranges of sea ice variability, directly comparing the SPS values in the two hemispheres is unfair. We thus normalize the SPS values by dividing them by the SPS of a trend climatology forecast for each hemisphere. The normalized SPS is referred to as nSPS.





## 3 Results and Discussion

### 3.1 Sea Ice ICs

We first compare the two sea ice ICs. Section 3.1.1 evaluates SIE and SIC, and section 3.1.2 compares SIV. Note that the SIC and SIE fields are evaluated against the NSIDC observations, while the SIV fields are only compared against the GIOMAS and BSOSE datasets for reference.

### 3.1.1 SIC and SIE

Figure 1 shows the seasonal cycle of the regional and pan-Antarctic SIE from the two ICs and the NSIDC observations. SPEAR_IC underestimates summer SIE in general, especially in the West Pacific and Ross sectors, but it overestimates SIE in other seasons across the pan Antarctic. By assimilating SIC, SICDA_IC reduces the biases in all months. The largest bias reduction happens in the Weddell and A&B sectors, where most of the positive biases in winter and spring are corrected. The magnitudes of the positive biases in winter and spring SIE in the Indian and West Pacific sectors are also reduced noticeably. The positive biases in autumn are less affected in all the sectors. The summertime negative biases have not been reduced as substantially, except in the West Pacific sector. SICDA_IC also has higher detrended correlation with the observed SIE (Figure S1), showing that it simulates better interannual variability of regional Antarctic SIE.

Figure 2 shows the spatial pattern of SIC biases in four seasons. SPEAR_IC has negative biases across the Antarctic Ocean in summer (January-February-March), except the positive bias in the East Weddell Sea (Figure 2a), which offset the negative bias in the West Weddell Sea, so the biases don't show up in the Weddell SIE in Figure 1a. The bias sign changes in autumn (April-May-June; Figure 2b) when positive biases dominate the variability zone throughout the rest of the year (Figures 2c and d), while a negative bias emerges in winter (July-August-September; Figure 2c) and lasts through spring (October-November-December; Figure 2d) in the interior Eastern Weddell and West Indian sectors. The Ross sector also has an expansive negative bias in spring, which cancels the positive biases elsewhere and results in a less biased regional SIE in the Ross sector as shown in Figure 1d. The negative biases (in winter and spring in the Weddell sector and spring in the Ross sector) are located in the regions where open-water polynyas tend to happen, suggesting that the polynyas simulated in SPEAR_IC might be too active or intense. SICDA_IC decreases the magnitude of both positive and negative biases across the Antarctic (Figures 2i–l). Notably, the wintertime and springtime positive biases in the Weddell and Ross sectors in SPEAR_IC are almost gone in SICDA_IC (Figures 2g and h). The impact of SIC DA is much smaller in summer and autumn. In summer, SICDA_IC reduces the negative biases in the West Weddell and West Pacific sectors, and the positive bias in the East Weddell Sector (Figure 2e). In autumn, SIC DA reduces the positive biases partly, but there are still sizable positive biases around the Antarctic (Figure 2f). There are also slight bias increases along the coastal Indian Sector, part of the coastal West Pacific in summer (Figure 2i) and spring (Figure 2l).

SICDA_IC shows predominantly better interannual variability of SIC than SPEAR_IC (Figure 3). The largest improvements are in spring both in the variability zone and the interior ice pack, followed by additional large improvements in winter in the interior, and summer in the variability zone, and smaller improvements in autumn. SICDA_IC also reduces total error almost



everywhere compared to SPEAR_IC as shown in the RMSE difference map (Figure 4). The RMSE reduction is mostly located in the ice variability zone, while it remains less impacted in the interior. One small exception is the RMSE increase that appears along the coast of the East Weddell in January and December, and along the coast of the West Pacific in December. Compared to the Arctic, the Antarctic has a much larger ice variability zone since it's not bounded by land. SICDA_IC has, on average, a smaller ACC increase (Figure S2) and RMSE decrease (Figure S3) in the ice variability zone in the Arctic than in the Antarctic. The maximum improvement from SIC DA seems to happen in different seasons in the two hemispheres. While the Antarctic sea ice benefits from SIC DA mostly in spring and winter, the Arctic sea ice shows the largest improvement in summer.

### 3.1.2 SIV

We compare SIV from the two ICs and two reference datasets, GIOMAS and BSOSE. Figure 5 shows the monthly mean SIV averaged over the common period (2008–2012) of the four data sets. There are significant discrepancies among the datasets. GIOMAS has the thickest mean SIV in all the regions throughout the year. SPEAR_IC, SICDA_IC, and BSOSE are closer to each other, and they mostly overlap within two standard deviations. BSOSE generally has larger mean SIV than SPEAR_IC except in the Indian and A&B sectors, where it has lower SIV in the winter months. SICDA_IC has the lowest mean SIV in almost all the regions in all months, with the only exception in the Ross Sea where SICDA_IC is nearly the same as SPEAR_IC. This homogeneous reduction in SIV seen in SICDA_IC from SPEAR_IC is consistent with the overall decrease in SIE as shown in Figure 1. We discussed in Zhang et al. (2021) that SIC and SIT usually change in the same direction due to SIC DA. Since our background model has a systematic positive bias in SIC/SIE, SIC DA tends to reduce SIC/SIE and, consequently, SIT/SIV. The same plot but over a longer record for GIOMAS and our ICs indicate that this feature is generic (Figure S3).

It is also interesting to note that the four datasets do not peak in the same month in the West Pacific, Ross, and A&B sectors. In the West Pacific sector, GIOMAS, SPEAR_IC, and SICDA_IC peak in September, while BSOSE peaks in October. In the Ross Sea, GIOMAS peaks in October, BSOSE peaks in September, and our two initial conditions peak in August. In the A&B sector, GIOMAS and SICDA_IC peak in September, SPEAR_IC peaks in October, and BSOSE peaks in November. For the pan-Antarctic SIV, BSOSE peaks in October, while the rest peak in September. A longer period suggests this difference in the SIV maximum month is typical in GIOMAS and the two ICs (Figure S4).

Figure 6 shows that all datasets agree that the majority of summer minimum ice is located in the West Weddell Sea. In summer, while SPEAR_IC, GIOMAS, and BSOSE have some ice thicker than 0.6m in the Ross and A&B sectors, SICDA_IC has no ice thicker than 0.05m in those regions. As Figure 5 shows, their discrepancy gets larger in the growing and peak seasons. GIOMAS has much thicker ice than the other datasets across the Antarctic Ocean (Figure 6c). SICDA_IC has the thinnest ice overall (Figure 6b). As sea ice grows, a thick sea ice band extending from the West Weddell Sea to the East Weddell Sea emerges in all datasets. This band is narrower in SPEAR_IC and expands northeasterly to the Indian Sector, which creates a hole in the West Indian Sector in spring. This feature is not seen in the other data sets. GIOMAS and BSOSE have generally thicker sea ice along the coast than the two ICs. Figure 6e highlights the SIV difference between the two ICs. SICDA_IC is overall thinner than SPEAR_IC along the coast in all seasons, with some exceptions. SICDA_IC has thicker ice in the West Weddell in summer and in the interior Weddell and Indian Sectors in winter and spring. It also has thicker ice in the Ross Sea





in all seasons. These are also regions where SICDA_IC reduces the negative SIC biases, as shown in Figure 2. A longer period shows similar features for GIOMAS, SPEAR_IC, and SICDA_IC (Figure S5).

## 3.2 Reforecast Skill

This section evaluates the skill differences between the two reforecast experiment suites: SPEAR and SICDA. We use the
225 damped persistence (DP) forecasts as a benchmark. We also compare the skill differences in the Antarctic to those in the Arctic to assess whether and how SIC DA influences forecast skill differently in the two poles. We use the detrended ACC and RMSE as skill metrics to evaluate the performance of their grid-cell level SIC predictions in Section 3.2.1, and nSPS to evaluate their probabilistic predictions of the sea ice edge position in Section 3.2.2. Finally, we discuss the skill degradations found in SICDA in Section 3.2.3.

### 230 3.2.1 Grid-cell level SIC

Figure 7 shows that SICDA has increased the 45-day mean forecast skill in terms of the detrended SIC ACC for all the initialization months. The most considerable improvement is in spring (October to December). The skill improvement is also noticeable in winter (July to September) and summer (January to March) but negligible in autumn (April to June). However, in the Arctic, the SIC skill improvement is most significant in summer (July to September), moderate in spring (April to June),
and minor in other seasons (Figure S6).

We take an area-weighted average of the detrended ACC in the ice variability zones in both poles for each initialization season and also annual mean in Figure 8. The DP skill decays slowest in the summer-initialized forecast in both hemispheres, followed by spring, winter, and autumn. This suggests that the SIC anomaly persists longer and thus provides more predictability in summer and spring at subseasonal time scales than in winter and autumn. Figure 8 shows that the DP forecast skill for
the Antarctic sea ice is better than that for the Arctic in spring and summer, similar in winter and autumn, thereby giving an overall better skill for the Antarctic than the Arctic. It indicates that the SIC anomaly is more persistent in the Antarctic than in the Arctic at subseasonal time scales.

Both reforecast experiments lose to the DP in the beginning but are able to beat the DP within the first month. SICDA performs significantly better than SPEAR at short lead times in both hemispheres, and its advantage decreases gradually with
245 forecast days. The initial skill gaps between the two reforecast experiments are more prominent in the Antarctic than in the Arctic for all seasons except summer (compared solid to dashed lines). Figure 8 also shows that SPEAR is generally less skillful in the Antarctic than the Arctic, suggesting a larger room for correction in the Antarctic. In contrast, SICDA shows slightly better skill in the Antarctic than in the Arctic in the first two weeks. This confirms that the sea ice IC is improved more in the Antarctic than in the Arctic by assimilating SIC observations.



### 3.2.2 Sea Ice Edge Predictions

We use the skill metric nSPS to assess the probabilistic prediction skill of the sea ice edge from the two reforecast experiments and how they compare to the DP forecast (Figure 9). A smaller/larger than one nSPS value indicates that the forecast is better/worse than the trend climatology. Since the DP forecast incorporates the anomaly persistence forecast in the short term and trend climatology in the longer term, its nSPS values should be smaller than one at short lead times and converge to one at longer lead times. And if a model reforecast has smaller nSPS values than the DP forecast, it's considered to have a skillful probabilistic prediction of sea ice edges. The DP forecast has slightly smaller nSPS values in the Antarctic than in the Arctic, except in winter (Fig. 9). This is consistent with its relative skill difference in the two hemispheres regarding the detrended SIC ACC discussed earlier.

SPEAR has nSPS values that are close to or higher than one in the winter and spring-initialized forecasts (Figs. 9 a and b), meaning it does not have skill in winter or spring in the Antarctic. Its SPS values do not increase much with forecast days, indicating a sizeable initial sea ice edge bias in its IC. Consequently, SICDA displays the most considerable skill improvement in these two seasons. Its nSPS values are significantly lower than those of SPEAR in the entire 45 days in the winter-initialized reforecast, and for about 40 days in the spring-initialized reforecast (see red dots in Figs. 9 a and b). SICDA starts to beat the DP forecast around day 15 in winter, but remains less skillful than the DP in spring. Both SPEAR and SICDA make skillful predictions of the sea ice edge for summer and autumn. SPEAR passes the DP at ~day 20 in summer and ~day 5 in autumn, while SICDA passes DP at ~day 15 in summer and ~day 2 in autumn. SICDA has significantly better skill than SPEAR for about 12 days in summer and 8 days in autumn, which is a less prominent improvement than in winter and spring. Annually, SICDA performs significantly better than SPEAR for 35 days. While SPEAR fails to beat the DP within the 45 forecast days, SICDA outperforms the DP after day 10.

In the Arctic, SPEAR shows better skill in winter and spring than in the Antarctic. It beats the DP quickly in both poles (~2 days in the Arctic and ~6 days in the Antarctic). It remains less skillful than the DP in spring but ties with it by the end of the 45 forecast days. SICDA consistently outperforms SPEAR; its improvement is statistically significant for 10 days in winter and 27 days in spring, which lasts shorter than in the Antarctic. It also beats the DP after ~20 days in spring, while it always loses to the DP in the Antarctic. In summer and autumn, the nSPS values in the Arctic of SPEAR are very similar to those in the Antarctic. The skill improvement from SICDA is 95% statistically significant for 21 days in summer and not significant at all in autumn. Annually, both reforecasts show better skill in the Arctic than in the Antarctic relative to the DP forecast. The improvement from SIC DA lasts shorter in the Arctic (22 days) than in the Antarctic (35 days).

### 3.2.3 Degradations

Figure 10 shows the skill evolution in terms of pan-Antarctic averaged detrended SIC ACC for each initialization month. For most months, there is an initial improvement from SICDA over SPEAR, and the skill values gradually merge. The size of the initial gap and how long it lasts vary by season. However, in the January and December-initialized reforecasts, SICDA loses



to SPEAR after 20 forecast days. This degradation is not seen in the Arctic, where SICDA consistently merges with SPEAR after an initial improvement (Figure S7). Figure 7 indicates that the degradation happens mainly along the coast.

We compare the RMSE of SIC from SICDA and SPEAR in the Antarctic (Figure 11) and find that SICDA reduces RMSE in
almost all the initialization months, but not in January or December. A similar skill degradation associated with SIC DA does not occur in the Arctic (Figure S2). The RMSE increase also happens around coastal Antarctica in both initialization months (Figures 4 a and l). Since the degradation is more prominent in the December-initialized reforecast, we focus on analyzing the results in December. Figure 12 shows that SICDA has a much higher detrended ACC and smaller bias on the first forecast day. The outer ring of positive bias in SPEAR is muted in SICDA. This confirms the success of SIC DA in improving initial
conditions. However, SICDA has slightly more negative bias along the coast in the West Pacific sector than SPEAR (Figures 12g and j). The SPEAR positive bias in the coastal Weddell sector is flipped to a negative bias in SICDA. Although SICDA shows higher detrended ACC in those regions, its RMSE is slightly larger than SPEAR on the first day (not shown), which indicates that the RMSE is dominated by bias. This negative bias along the coast in SICDA grows more extensive with forecast time. The pan-Antarctic averaged absolute bias of SIC in SICDA is equal to SPEAR around day 20 (Figures 12h and k). While
the bias along the West Pacific coast does not grow much with lead time, the bias continues to expand in the Weddell and A&B sectors and becomes sizable at day 40 (Figure 12l). In the meantime, the detrended ACC decays with forecast time, and SICDA loses skill faster than SPEAR, which is likely affected by its faster-growing bias. The co-location of an initial negative SIC bias and the faster decay in skill for SICDA suggests that the SIC-based sea ice albedo plays a role in exacerbating the low SIC bias. Also, as we showed in Section 3.1.2, SICDA_IC has thinner SIV than SPEAR_IC along the West Pacific coast.
It suggests that the SIT-based sea ice albedo feedback is also essential.

We also notice that SICDA develops positive SIC biases in the interior of the Ross and Weddell sectors during the first 45 forecast days (for example, on Day 20 and 40 shown in Figures 12k and l), where no noticeable SIC biases existed on the first day (Figure 12j) or in its IC. The interior of Ross and Weddell sectors are the regions where SPEAR has negative biases on the first day (Figure 12g) and gradually forecasts positive biases later on (Figures 12h and i). It indicates that the sea ice melt
rates in these two regions are too slow in the SPEAR system. Thus, although SICDA is initialized from a more neutral (less negative) SIC initial condition, the intrinsic slow melt rate will gradually cause more positive SIC bias in these two regions than SPEAR does as it simply has more sea ice to melt. Plus, SICDA_IC also has thicker sea ice in those regions, as shown in Figure 6e, the melt rate in SICDA maybe even slower than in SPEAR, which helps produce a more positive SIC bias in the interior Ross and Weddell sectors.

Figure S8 displays the difference in the forecasted December monthly-mean states of SIC, SIV, net downward shortwave radiation (SW), and SST in the first month of the December-initialized forecasts. SICDA has up to 40% lower SIC (Figure S8a) and 0.5m thinner SIV (Figure S8b) than SPEAR along the coastal Antarctic, which is more prominent and expansive than their difference in ICs (c.f Fig. 2). In the meantime, SICDA has up to 50 $w/m^2$ more SW (Figure S8c), and 0.8° warmer SST (Figure S8d) than SPEAR, suggesting that the combination of SIC-based and SIV-based albedo feedback exacerbates the
initial SIC difference by absorbing more solar radiation, warming up the SST, inducing more bottom melt, and in turn resulting





in even less SIC and SIV in SICDA than SPEAR. At the same time, SICDA has more expansive and thicker ice in the interior Ross and Weddell sectors, which reduces SW, cools SST, and creates a condition favoring higher SIC and SIV.

## 4    Conclusions

We have evaluated the Antarctic sea ice concentration (SIC), extent (SIE), and volume (SIV) of a sea ice concentration (SIC)
data assimilation (DA) product (SICDA_IC) generated using the sea ice DA framework of GFDL MOM6/SIS2 and Data Assimilation Research Testbed. We compared the sea ice states against the standard sea ice initial condition (IC) for the GFDL SPEAR_MED seasonal prediction system (SPEAR_IC) to examine the benefits of SIC DA.

We find that SICDA_IC largely ameliorates the positive biases in winter SIE and the negative biases in summer SIE in SPEAR_IC. SICDA_IC also notably improves the interannual variability of regional SIE. By evaluating the grid-cell level
SIC, we find that SPEAR_IC has negative SIC biases along the coast and positive SIC biases in the East Weddell sector in summer, a ring of positive bias in the marginal ice zone persists through the rest of the year, negative biases emerging in the interior Weddell sector in winter which expands in spring, and negative biases in the interior Ross sector in spring. The interior Weddell and Ross sectors are also where open-water polynyas happen. It suggests that the polynyas in these two sectors are likely too active in the SPEAR system. SICDA_IC reduces the bias across the Antarctic, marked by the reduction of the ring of
positive bias that persists from from autumn to spring and the negative bias in the polynya active regions in winter and autumn. A slight bias increase is seen along the coastal Antarctic in summer and spring, where the negative bias in the West Pacific sector in SPEAR_IC becomes slightly more negative in SICDA_IC, and the positive bias in the East Weddell and Indian sectors in SPEAR_IC becomes slightly positive in SICDA_IC.

SICDA_IC improves the interannual variability of SIC and reduces the RMSE of SIC almost everywhere in all seasons.
The seasonality of skill improvement differs in the Antarctic from in the Arctic. Our previous Arctic sea ice study showed that the improvement from SIC DA is largest in summer, followed by spring and winter, and negligible in autumn. While in the Antarctic, we find that the largest improvement happens in spring and winter, followed by summer, and is the smallest in autumn.

We conducted two reforecast experiments, SPEAR and SICDA, initialized from SPEAR_IC and SICDA_IC, respectively.
We evaluated the detrended ACC and RMSE of the grid-cell level SIC and found an overall improvement in SICDA at sub-seasonal time scales. The seasonality of skill improvement regarding the detrended SIC ACC is similar to that of its IC. The improvement is largest in winter, followed by spring, summer, and autumn. We compare the results in the Antarctic to that in the Arctic and show that the magnitude of skill improvement is larger in the Antarctic than in the Arctic.

The assessment of the probabilistic sea ice edge prediction skill shows that SICDA performs significantly better than SPEAR
in both hemispheres in all seasons. Contrasting the skill in the two hemispheres, we found that SICDA has a larger initial improvement in the Antarctic than in the Arctic. A bootstrap significance test also shows that SICDA has significantly better forecast skill than SPEAR for more days in the Antarctic (35 days) than in the Arctic (22 days) based on nSPS values averaged over the 12 initialization months. The nSPS values from all experiments have a strong seasonality. In the Antarctic, SPEAR





has the lowest skill in winter and spring-initialized reforecasts. In these seasons, SPEAR has considerable initial errors, which

do not increase with forecast days very much. SPEAR does not beat DP throughout the 45 forecast days in these seasons. In contrast, SICDA has the largest improvement in winter and spring-initialized reforecasts. It indicates that SIC DA is most useful when SPEAR suffers from significant initial errors. In autumn, the nSPS from the DP reaches the maximum value much faster than in other seasons, suggesting short-lasting sea ice edge memory in this season. SPEAR surpasses the DP shortly, leaving SICDA little room for improvement.

Comparing our sea ice edge prediction skill analysis to Zampieri et al. (2019) and Gao et al. (2024), we show that the SICDA performs close to ECMWF in both poles, which shows higher sub-seasonal prediction skills than the benchmark (a damped persistence forecast) after the initial loss. Without SIC DA, SPEAR does not beat the benchmark in the Antarctic and it takes longer for SPEAR to outperform the benchmark in the Arctic.

We found a small skill reduction in the January and December-initialized reforecast experiments from SICDA. We analyze

the December-initialized reforecasts to diagnose the cause of the degradation. By evaluating the daily skill evolution of SICDA and SPEAR, we find that SICDA has overall better initial skill than SPEAR; however, it starts to lose to SPEAR around Day 20. The skill reduction happens mainly in the coastal Antarctic and interior Weddell and Ross sectors. In the coastal Antarctic, SICDA_IC has lower SIC and thinner SIV than SPEAR_IC, which triggers a combination of SIC-based and SIV-based ice albedo feedback that lowers the sea ice albedo, increases the incoming solar radiation, warms the SST, and melts more sea ice,

which contributes to the negative biases in SICDA. In the interior Weddell and Ross sectors, SICDA_IC has higher SIC and thicker SIV than SPEAR_IC. Although SICDA has more realistic SIC IC in these two regions, a more expansive and thick sea ice pack further slows down the already too-slow melt rate in the interior Weddell and Ross sectors predicted by the SPEAR system, exacerbating the positive bias after 20 days. While we don't have ground truth observations to judge whether the SIV field is adjusted in the right direction via SIC DA, the SIC and SIV fields in SICDA may not be compatible in this particular

configuration of the SPEAR system. It suggests that we must be careful in adjusting multiple highly related fields through DA as it may cause an imbalance among model states that leads to skill degradation.

In summary, we demonstrate that SIC DA successfully generates a more accurate sea ice IC in the Antarctic. The improvement in grid-cell SIC and sea ice edge predictions from SIC DA is more prominent in the Antarctic than in the Arctic, partly due to the larger room for correction in the Antarctic sea ice in the SPEAR system, especially in spring and winter. As a result,

the reforecasts initialized from the SIC DA product make significantly better subseasonal forecasts of grid-cell level SIC and sea ice edges in all initialization months; the reforecast skill improvement in the Antarctic is also more significant than in the Arctic. A slight degradation is spotted in the December- and January-initialized reforecast experiments. It is believed to be related to the imbalance between the SIC and SIV fields due to SIC DA. Future work on improving the model physics and DA techniques is needed to generate more compatible sea ice fields via DA.

*Data availability.* The initialization data and reforecast experiment results are shared at Zhang (2024).



*Author contributions.* YZ, MB, and MW conceptualized the study and designed the experiments. YZ prepared the SICDA_IC data set, conducted the SICDA reforecast experiments and analyzed the model results. YZ, MB, MW, WG, and BH discussed results on a weekly basis. LJ prepared the atmosphere/land/sea ice data for SPEAR_IC and conducted the SPEAR reforecast experiments. FL prepared the ocean initial conditions for SPEAR_IC. YZ wrote the first manuscript with the help of MB, MW, WG, and BH. All coauthors contributed to the editing.

*Competing interests.* The authors declare that they have no conflict of interest.

*Acknowledgements.* Yong-Fei Zhang received award NA18OAR4320123 under the Cooperative Institute for Modeling the Earth System (CIMES) at Princeton University, and the National Oceanic and Atmospheric Administration, U.S. Department of Commerce. We thank Liping Zhang and Theresa Cordero for providing constructive suggestions during the GFDL internal review.



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





**Figure 1.** 26-year (1992–2017) mean seasonal cycle of regional and pan-Antarctic SIE for SPEAR_IC (blue), SICDA_IC (red), and the NSIDC observations (black). Each error bar represents two standard deviations of regional SIE calculated over the years.





**Figure 2.** SIC Bias for each season for SPEAR_IC (a-d), and SICDA_IC and (e—h), and their absolute bias difference (ABD; SICDA_IC - SPEAR_IC; i–l). The biases are calculated against NSIDC SIC observations from 1992–2017.





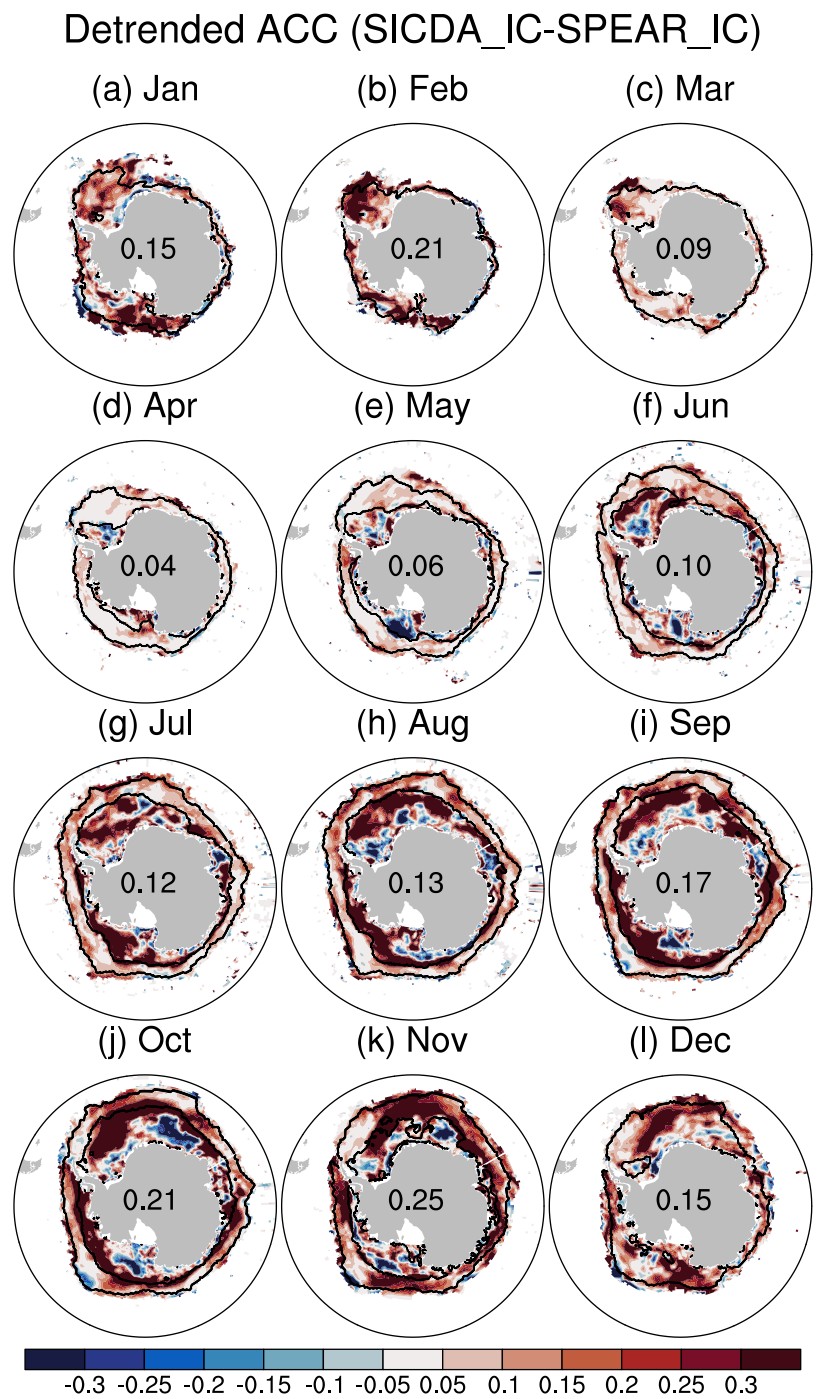

**Figure 3.** Difference in detrended SIC ACC between the two ICs (SICDA_IC - SPEAR_IC) for each month. The black lines on each map represent the observed 10% SIC variability zone. The numbers on each map are the area-weighted average of the detrended ACC difference within the 10% SIC variability zone.





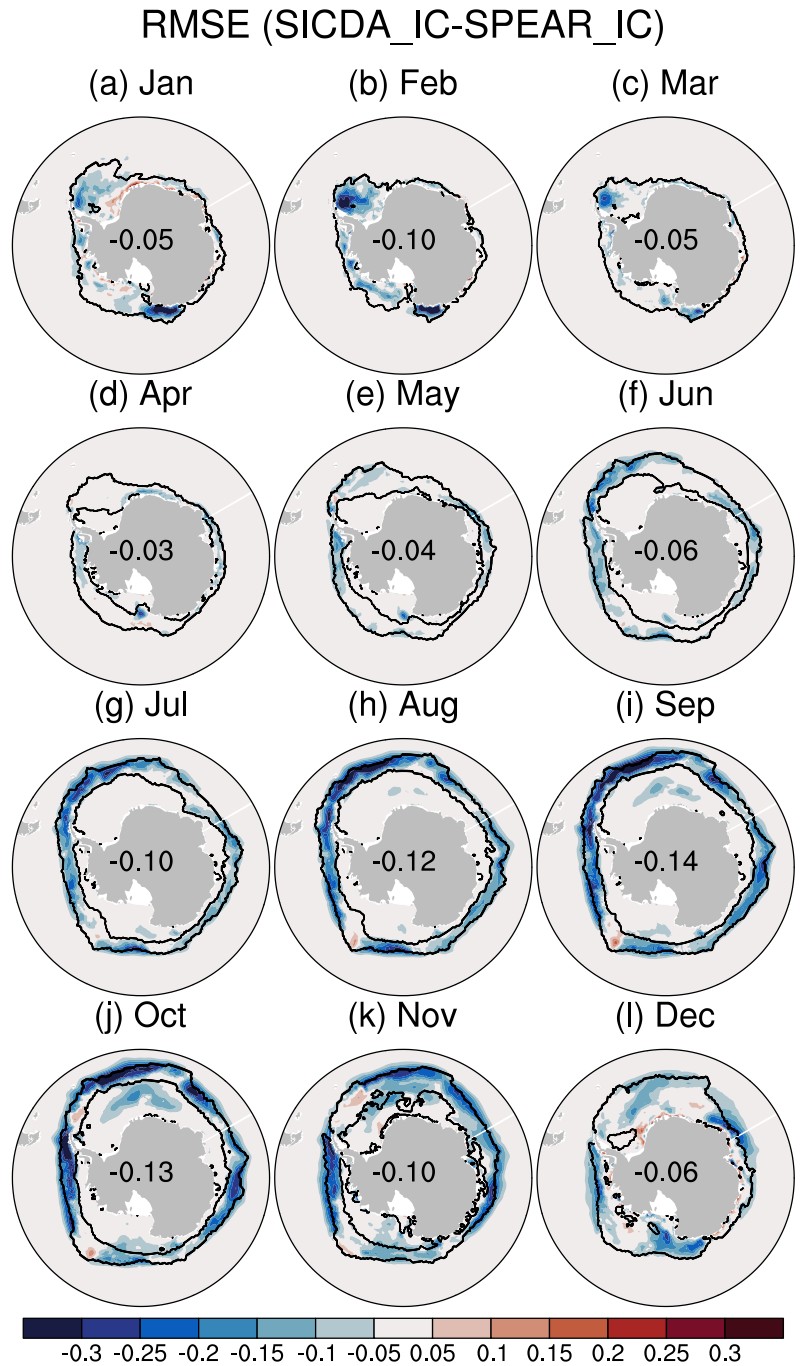

**Figure 4.** Difference in SIC RMSE between the two ICs (SICDA_IC - SPEAR_IC) for each month. The black lines on each map represent the observed 10% SIC variability zone. The numbers on each map are the area-weighted average of the RMSE difference within the SIC variability zone.





**Figure 5.** 5-year mean seasonal cycle of the regional and pan-Antarctic SIV from SPEAR_IC (blue), SICDA_IC (red), GIOMAS (black), and BSOSE (gray).The values are calculated over their overlapping years, 2008–2012. Each error bar represents two standard deviations of SIV calculated across the years.





**Figure 6.** Spatial map of the 5-year mean SIV for each season from the four data sets: (a) SPEAR_IC, (b) SICDA_IC, (c) GIOMAS, and (d) BSOSE, and (e) the difference between SICDA_IC and SPEAR_IC.




**Figure 7.** Difference in the 45-forcast-day mean of the detrended SIC ACC between the two reforecast experiments (SICDA - SPEAR). The black lines on each map represent the observed 10% SIC interannual variability zone. The numbers on each map are the area-weighted average of the ACC difference within the SIC variability zone.



# Detrended SIC ACC (1992-2017)

**Figure 8.** Detrended SIC ACC averaged over the Antarctic ice variability zone as a function of forecast days for SPEAR (red dashed lines), SICDA (red solid lines), and DP (black lines), and over the Arctic ice variability zone for SPEAR (blue dashed lines), SICDA (blue solid lines), and DP (gray lines). The Detrended ACC values are calculated for each initialization month and then averaged over the (a) winter months, (b) spring months, (c) summer months, (d) autumn months, and (e) all 12 months. Each red dot on the bottom indicates that the detrended ACC of SICDA is significantly higher than that of SPEAR, and red circle the detrended SIC ACC of SIDA is 95% significantly worse than that of SPEAR for the corresponding forecast day in the Antarctic. The blue dots and circles are for the Arctic, respectively.




**Figure 9.** Normalized SPS calculated over the Antarctic as a function of forecast days for SPEAR (red dashed lines), SICDA (red solid lines) and DP (black lines), and over the Arctic for SPEAR (blue dashed lines), SICDA (blue solid lines) and DP (gray lines). The SPS values are normalized by the SPS values of the trend climatology forecast for each hemisphere. The normalized SPS values are calculated for each initialization month and then averaged over the (a) winter months, (b) spring months, (c) summer months, (d) autumn months, and (e) all 12 months. Each red dot on the bottom indicates that the SPS value from SICDA is lower than that of SPEAR at the 95% significant level. The blue dots are for the Arctic.



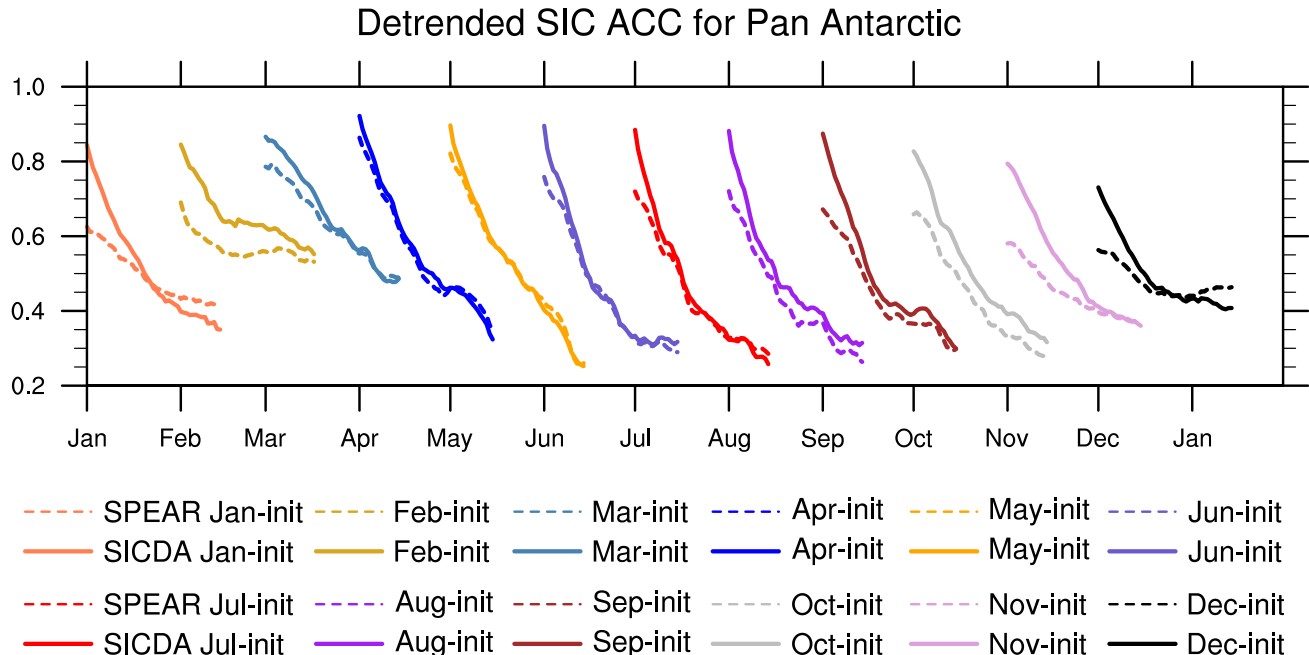

**Figure 10.** Detrended SIC ACC averaged over the pan-Antarctic ice variability zone as a function of forecast days from the 12 initialization months for SPEAR (red dashed lines), SICDA (red solid lines), and DP (black lines), and over the pan-Arctic ice variability zone for SPEAR (blue dashed lines), SICDA (blue solid lines), and DP (gray lines). The ice variability zones are regions where the observed SIC standard deviation in the experimental period is larger than 10%.





**Figure 11.** 45-forecast-day mean of the SIC RMSE difference between the two reforecast experiments (SICDA - SPEAR). The black lines on each map represent the 10% SIC interannual variability zone. The numbers on each map are the area-weighted average of the RMSE difference within the 10% SIC variability zone.







**Figure 12.** SIC detrended ACC, and bias on the forecast days of 1, 20, and 40 from the December-initialized reforecast experiments for SPEAR and SICDA. The skills are calculated using daily data from 1992–2017.