# Peer review of "Subseasonal Forecast Improvements from Sea Ice Concentration Data Assimilation in the Antarctic"

_EGUsphere, 2025_

## Author Comment (AC1)

**Response to RC2**

We thank the reviewer for carefully reading our manuscript and providing constructive comments that helped improve it. We have carefully considered all the comments and provided detailed, point-by-point responses. The reviewer's comments are copied below in regular font and our responses are shown in bold.

In this study, the authors investigated the impact of assimilating SIC data on improving sea ice initial conditions (ICs) and prediction skill. Their results demonstrated that SIC assimilation generally improves both the IC and subsequent predictions, although the improvement exhibits strong regional and seasonal dependence. A slight decrease in skill occurs in the Weddell Sea and the Ross Sea for prediction initialized from December and January. The possible reasons accounting for this were also discussed. In addition, the authors compared the influence of SIC assimilation between two polar regions and concluded that SIC assimilation has a larger impact in the Antarctic than in the Arctic, which is of broad scientific interest to many researchers.

Overall, I found this paper is well organized, clearly expressed and addresses a cutting-edge research topic. I only have some minor comments and suggestions listed as below. I recommend minor revision for this manuscript.

**Specific Comments:**

1) Line 45, a recent study also demonstrated that SIT is a strong source of predictability for summer sea ice in the Weddell Sea, and it can be well constrained through atmospheric initialization.

Thanks for providing the additional reference. We've added the reference into our introduction in Line 46. The text is also copied below.

"Bushuk et al. (2021) and Xiu et al. (2025) found that SIT is a strong source for summer sea ice predictions in the Weddell Sea, which can be well constrained by realistic atmospheric forcings."

Reference: Xiu et al. Impact of Ocean, Sea Ice or Atmosphere Initialization on Seasonal Prediction of Regional Antarctic Sea Ice. Journal of Advances in Modeling Earth Systems 17, e2024MS004382 (2025)

2) Line 60, please correct "C2S" to "C3S".

We thank the reviewer for pointing out the typo. We've corrected it in the manuscript.

3) Section 2.2, please consider to add some descriptions on how other variables (e.g., SIT) adjust in response to SIC DA since SIT is later used for analysis.

Thanks for the suggestion. We added some descriptions in Section 2.2. The text is copied below.

"The SIC of each category (5 thickness categories in total) is the only state variable that's directly updated by DA. The ice and snow thickness of each category remain unchanged, while their aggregate thicknesses are adjusted by the update in the concentration of all categories."

4) Line 103, (Bushuk et al., 2021).

We've corrected the format. Thanks for the edit.

5) Section 2.3, what is the frequency of the hindcast?

We added a table to list the configuration of the two hindcasts (table 1) and modified Section 2.3 accordingly. Please see the modified text below.

"Each reforecast experiment consists of 15 ensemble members, covering the period 1992--2017. The reforecasts are initialized on the first day of each month and integrated for one year. Only the first 45 days are analyzed in this study as we focus on the subseasonal time scale of their prediction skills. "

6) Line 123, it could be misleading to claim that GIOMAS can reasonably represent the Antarctic sea ice thickness climatology, as it shows large discrepancy with the satellite-based observation (e.g., Figure 3 in Liao et al. (2022)).

We agree that GIMAS suffers from biases. We've rephrased the sentence in Line 129. The modified text is copied below.

"Previous studies (e.g., Liao et al., 2021; Shi et al., 2020) evaluated it against satellite retrievals, shipping and airborne observations and concluded that the GIOMAS SIT field can reasonably represent the interannual variability, and trends of Antarctic sea ice, although it suffers from biases. It was found to underestimate SIT, especially in the deformed ice zone, for example, the northwestern Weddell Sea."

7) Lines 140-141, just for clarification: do you first compute the ensemble-mean SIC and then calculate SIE from the ensemble mean?

Yes we compute the ensemble-mean SIC and then calculate SIE.

8) Lines 165-166 and lines 180-185, do you have any hypotheses on the seasonally dependent improvements from DA?

We have some speculations on the seasonal differences in the DA performance. The sea ice DA tends to be more effective when there are larger errors to be corrected

from the first place. Figure 2 shows that the SIE of SPEAR\_IC has relatively small negative bias in summer compared to the observation and much larger positive errors in winter. Hence there's more room for correction in the winter and spring. The similar contrast is seen in the spatial map of SIC bias in Figure 3 as well. The ice variability zone is much larger in the winter and spring seasons, where the model tends to have more uncertainty and larger bias.

9) Lines 221, I think the 'negative SIC biases' only hold for the spring, while in other seasons, the positive biases dominate (Figure 2).

Yes we agree that the negative SIC biases only appear in limited regions and times. We've rephrased the sentence in Line 225 to be more rigorous. Please find the text copied below.

"It also has thicker ice in the Ross Sea in all seasons. These are also regions where SICDA\_IC reduces the negative SIC biases, e.g., the Weddell sector in winter and spring and the Ross sector in spring (Figures 3g and h)"

10) Lines 238-239, Figure 1 suggests that some regions are ice free in summer. Considering this, where does the SIC anomaly in the summer originate, especially for the western Antarctic? Can you add a spatial pattern of detrended ACC to support this claim?

We agree that Figure 1 (Figure 2 in the updated manuscript) suggests some regions are ice free in summer, but there's still ice lingering, even in the western Antarctic. The following plot shows the detrended ACC for the winter months. It is true that the variability zone is mostly located in the West Antarctic: Weddell, A&B, and Ross. In the East Antarctic, there's still a narrow band of variability zone along the coast. The ice variability zones are also highlighted in Figure 3 (black lines).

11) Lines 241-242, It is somehow unexpected that the Damped Persistence is weaker in the Arctic than in the Antarctic because the Arctic SIT is thicker than Antarctic counterpart overall, as also mentioned in Lines 38-42. Is this conclusion strongly dependent on the assessment metric used?

The skill we chose is the detrended ACC of grid-cell SIC averaged over the ice variability zone. We believe it is a more informative matric than the total SIE which may cancel out errors spatially. Previous studies have focused on analyzing the total or regional SIE, but very few studies have looked at the persistence of grid cell SIC. Another metric nSPS (Figure 9) also suggests that the probabilistic prediction skill of ice edge position also decays slightly slower in the Antarctic than in the Arctic. This is also seen in Figure 1 of Zampieri et al (2019).

12) Lines 247-248, Please specify the season and lead times more precisely. I think it should write as 'Figure 8 also shows that SPEAR is generally less skillful in the Antarctic than the Arctic in autumn and winter, suggesting a larger room for correction in the Antarctic. In contrast, SICDA shows slightly better skill in the Antarctic than in the Arctic in the first two weeks in winter and summer.'

**Thanks for the suggestion. We have edited the text as the reviewer suggested.**

13) Lines 297-299, The statement "The co-location of an initial negative SIC bias and the faster decay in skill for SICDA suggests that the SIC-based sea ice albedo plays a role in exacerbating the low SIC bias" maybe need to be further clairfied. Specifically, what is the role of model error v.s. initial error in degrading prediction skill. For example, I notice that the SPEAR has no negative bias along the Weddell Sea coast (Figure 12g), but its prediction bias turns out negative. This obviously can't be solely explained by the ice albedo

feedback. So I'm wondering how model error versus initial error contributes to skill degradation?

We thank the reviewer for the comment. As the reviewer pointed out, there's a development of negative SIC bias in the West Weddell Sea coast in both SPEAR and SICDA (seen in Days 20 and 40). SPEAR starts with neutral SIC condition and slowly develops the negative bias in this region, which indicates an intrinsic model bias. The mass budget analysis of the SPEAR system in the Weddell Sea (Figure 14c in Bushuk et al 2021) shows that the decrease of sea ice in summer is dominated by sea ice melt. Hence the negative bias in this region suggests that SPEAR tends to melt too fast in this particular region in summer. And we know that SICDA starts with a thinner SIT (Figure 6) in West Weddell coast, which explains why SICDA melts even faster and develops a more negative bias. We certainly didn't mean the IC error is the only reason for degraded skill, we agree with the reviewer it's a combination of IC and model errors. We didn't elaborate on this West Weddell coast bias because we didn't see skill degradation in this region in detrended ACC or RMSE. The SIC bias does seem larger in the SICDA Dec-initialized run, which suggests that the slight improvement in ACC cancels out the slight increase in bias and leads to negligible difference in their RMSE difference.

We actually discussed the role of model intrinsic bias in the next paragraph in the manuscript, although in a different scenario, e.g., SPEAR melts too slowly compared to the observations in the Ross sector, which compensates for its initial negative bias and results in less biased SIC condition in the forecasts (Line 309–317).

It is out of our scope to quantify the contributions of IC and model intrinsic errors, but we added the West Weddell coast case in the manuscript to emphasize that model intrinsic bias also plays a role in the skill differences between the two reforecasts (Line 318–321).

"The interplay between model intrinsic bias and the exacerbation from ice albedo feedback also manifests in the West Weddell coast. SPEAR starts with close-to-observation SIC in this region (Figure 12g) and develops negative SIC bias with time (Figures 12h and i), which indicates that sea ice melts too fast in SPEAR in December. With close-to-observation SIC to start with also, SICDA has thinner SIT IC in the West Weddell coast than SPEAR, hence the sea ice in SICDA melts even faster and shows worse negative SIC bias by Day 40 (Figure 12i)."

Comments on Figures:

Figure 5, please add the units to y-axis label

Label added, thanks!

Figure 8, SICDA

We thank the reviewer for catching the typo. It's been corrected.

---

## Author Comment (AC2)

**Response to RC1**

We thank reviewer 1 for providing valuable comments and constructive suggestions on our manuscript. We have responded to the reviewer's comments point by point. The comments are pasted below in regular font and our responses are in bold.

RC1: This paper deals with applied problem of subseasonal prediction of Antarctic sea ice concentration/extent/volume. It looks in detail at two reforecast experiments, one with and one without data assimliation, and compares this to an observational sea ice record (plus a few other reference datasets). Results are discussed in terms of seasons and sectors of the Southern Ocean. A comparison to similar existing studies for the Arctic show a few interesting differences. For example, that sea ice predictability as a function of initialization season is not the same in the Antarctic as it is in the Arctic.

I don't know enough about the immediately adjacent literature to comment on the novelty or need for this study, but the paper seems solid. With a few exceptions (my comments below), the paper is organized well, the writing is easy to follow, the figures are clear, and the length seems appropriate.

**Detailed comments:**

1. Defining the two main datasets. This paper is focused on SPEAR vs SICDA, but this isn't made explicit early enough. It only started to become clear as I made my way through Section 2. The last paragraph of Section 1 hints at this comparison being important. And the end of Section 2.3 reiterates it a bit. But I'd suggest being more explicit. One option would be to move a couple of sentences from end of Sections 1 and 2.3 to the start of Section 2 (before Section 2.1 even). Make it very clear in one place. Another option would be to add a Table or Figure that summarizes Section 2 (what models will be used, what quantities will be compared, etc). Although it wouldn't add anything new, this Table/Figure would be a quick way for the reader to refer back and check the details quickly.

Thanks for the suggestion. We agree that having a table that lists the components and configurations of the two experiments will make it clearer to readers. We have added a table (Table 1 in the manuscript) per your suggestion.

- 2. Add sector lines to all map figures. Figures 2, 3, 4, 6, 7, 11, and 12 would benefit from having five lines radiating out from the south pole that separate the maps into the five sectors defined in Section 2.6. To make sense of Section 3, I found that I had to draw these lines on myself (at least onto the top left map in each figure). Consider labeling the sectors in Figure 2a, then just repeating the lines thereafter.
  - Thank you for the thoughtful suggestion. While adding lines in each plot may be virtually overshelming, we agree that providing a spatial map showing the definitions of all regions would be helpful for readers. We have therefore added Figure 1, which illustrates the boundaries of all regions referenced in the paper.

3. The paper uses a small number of acronyms, but uses them a lot. I'm not sure it's a problem that has a good solution, but it does make a lot of the sentences stilted. A couple of egregious cases are the definition of EAFK on L97 (which is never subsequently used) and DART on L67 and L96 (which is only used twice, but defined in full both times). Also, avoid acronyms in headings (subsections of 3.1). Just write out each phrase in full.

We thank the reviewer for pointing out the errors we made in the text. We've deleted the acronyms that are not used later and written out names in full in the headings.

4. Vague quantification in the Abstract: There is an overuse of vague adjectives in this part, rather than concrete numbers. Examples include 'significantly' (L5), 'considerable' (L6), 'mostly' (L6), 'improved the most' vs 'minor differences' (L11), 'more significant' (L12). Since many readers only interact with papers via the abstract, make sure to include at least some key metrics.

Thanks for the comments. We edited the abstract to include the key metrics used in the paper and replaced a few words to make the language more clear.

This study evaluates the impact of sea ice concentration (SIC) data assimilation (DA) on subseasonal forecasts of Antarctic sea ice by comparing reforecast experiment suites initialized with and without SIC DA. The two initial conditions (ICs) are evaluated against NSIDC SIC observations from 1992 to 2017. Assimilating SIC reduces mean biases and root-mean-square-errors (RMSE) and enhances anomaly correlation coefficients (ACC) of SIC. The improvement in sea ice ICs is greater in the Antarctic than in the Arctic. After SIC DA, the sea ice thickness (SIT) field becomes mostly thinner except in the interior Weddell and Ross sectors. Results from reforecast experiments show that SIC DA improves the subseasonal forecast skill of Antarctic SIC, as indicated by enhanced detrended ACC, in nearly all initialization months except December and January, when the initial improvement is quickly overtaken by a bias likely linked to the thin SIT bias. SIC DA also improves the probabilistic prediction of the sea ice edge position, as measured by the Spatial Probability Score (SPS) at subseasonal time scales. The forecast skill improvement is largest in spring, followed by winter and summer, and shows minor differences in autumn. Consistent with the IC improvement, the reforecast skill gain associated with SIC DA is more pronounced in the Antarctic than the Arctic. Our study demonstrates the critical role of SIC DA in improving subseasonal prediction of Antarctic sea ice.

- 5. There is also some vague quantification in other parts as well:
  - 'skillful predictions' at L48 (how skillful?)
  - "Skillful" meaning the model predictions exceed referenced statistical predictions, which is an anomaly persistent forecast in the study of Bushuk et al. (2021).
  - 'improved' at L49 (by how much?)

- 'outperformed' at L64 (by how much?)
- 'reasonably represent' at L122

We thank the reviewer for this helpful comment. The above statements noted as vague were in the Introduction, where our goal was to provide general context rather than detailed quantitative results. Because many of the cited studies do not report directly comparable metrics or regional values, we have revised the text to improve clarity and precision while keeping the discussion at an appropriate level of generality. Specifically, we rephrased several sentences to better reflect the key findings of previous work without implying unavailable quantitative detail.

- 'underestimate' at L124 (by how much)

Due to the lack of SIT observations in the Antarctic, these studies compared GIOMAS with different observation-based datasets, each having their own uncertainties and biases. The comparison also varies with season and space. So it is hard to provide a single number of how much bias it has. For example, Shi et al. (2021) evaluated four reanalysis datasets including GIOMAS in the Weddell Sea and found that GIOMAS has an average negative bias of -0.75m in autumn compared to IceSat-1, but didn't mention the mean bias in other seasons. Liao et al. (2022) showed that GIOMAS can have a significant negative bias of -1.99m at the upward-looking sonar (ULS) site 206, but didn't provide statistics for other sites. So we decide not to give a single number but refer the readers to read their papers in detail if they are interested in a specific region and time of year.

- 'noticeably' at L164

We deleted the vague word.

6. 15. 'Steady increase' is not the correct description of Antarctic sea ice from late 70s to 2015. It didn't decrease over this period, but it wasn't clearly increasing, and it certainly wasn't doing so steadily.

Thanks for the comment. We have changed "steady increase" to "an overall increasing trend".

7. 26, 30. Unclear what 'perfect' means here.

A "perfect model study" assumes models are free of biases and initial errors. It's a method commonly used in predictability studies to examine the upper bound of the predictability of a variable. We understand that the term might be unfamiliar outside of the community, hence we added some explanation when it first appears in the paper. The following sentence is added in Line 26.

"The value of initialization in Antarctic sea ice predictability has been assessed by previous studies using a perfect model approach, which assumes that the models and initializations are bias free."

8. 161. Define the months for each season earlier in Section 3. You do define these later (at L169), but at least for L161-167, I was left wondering exactly what, say, 'summer' corresponded to.

The four seasons are defined in Section 3.1, Line 164. The text is copied below.

"The twelve months are grouped into 4 seasons, defined as summer (January–March), aurum (April–June), winter (July–September), and spring (October–December)."

9. 194. Section 3.1.2 seems just as much about Thickness as it is about Volume. Perhaps this should be reflected in the heading.

We thank the reviewer for pointing out the confusion. We understand the two terms may be confusing to some readers. Sea ice thickness is the averaged thickness over the ice covered area, and sea ice volume is the averaged thickness per grid cell (defined in L132). So the two terms are used interchangeably sometimes, but we made sure to refer to the correct term for corresponding datasets. For example, both GIOMAS and BSOSE provide the averaged thickness per grid cell, i.e., SIV. Since we are comparing the two ICs with GIOMAS and BSOSE, we use the term "SIV" in Section 3.1.2 instead.

10. 197. 'thickest' seems like a weird adjective for describing sea ice volume.

It's rephrased to "largest".

11. 215. Replace 'from the West Weddell Sea to the East Weddell Sea' with just 'across the Weddell sector'?

Thanks for the edit. We rephrased the sentence in Line 221 to the following.

"As sea ice grows, a thick sea ice band extending from the west to east across the Weddell Sector emerges in all datasets."

12. 243. 'first month' seems a weird description. To me, 5–10 days would be a more appropriate description. Yes, some cases extend beyond 10 days, but a month is too much of an upper bound to be relevant for this sentence, right?

Thanks for pointing it out. We agree that "a month" is an understatement of the model skills compared to the reference reforecasts. We rephrased this sentence to reflect that both model predictions beat the reference within the first week in the annual-mean plot. Please see the text copied below.

"Both reforecast experiments lose to the DP in the beginning but are able to beat the DP within the first week annually (Figure 9e)"

13. 261. If I'm reading this sentence right, you're inferring that SICDA has the 'most considerable skill improvement' because it is better than SPEAR. But you're only comparing two cases. If so, the word 'most' is inappropriate as it should be reserved for comparing three or more cases.

Thanks for the comment. We meant SICDA has the "most considerable skill improvement" in the two seasons (winter and spring) when SPEAR doesn't show better skill than the reference forecasts.

14. 282. Not clear what 'This' refers to. Is it the underprediction in Dec–Jan? Or is it referring to the merging of skill values (from an earlier sentence).

The degradation refers to SICDA's worse skill than SPEAR after about 20 days. We've edited the text in Line 289 to clarify it. Please see the text copied below.

"The skill degradation of SICDA relative to SPEAR is not seen in the Arctic"

15. 303. Unclear how Figure 12j has 'no noticeable bias'. Are you implying that, say, the 0.07 average value is ≈0?

We meant there is no noticeable bias in the Ross and Weddell Sectors, following the sentence just before talking about the positive biases developed in these two Sectors. We've changed the sentence to make it clear. The Figure number is updated to Figure 13. "where no noticeable SIC biases existed on the first day (Figure 13j) or in its IC in these two Sectors".

16. 310-317. The last paragraph before the Conclusion only cites results that are shown in the Supplementary material. Why? One would think the last paragraph of a 'Results and Discussion' section is a good place for a key result. But relegating it to the Supplementary gives the opposite impression. Is there a 12-figure limit for this journal, so you decided putting the figure in the Supplementary was a good workaround? If so, it isn't.

Thank you for the suggestion. We respectfully disagree with the reviewer's concern about the potential negative impression. The "Results and Discussion" section begins with improvements in prediction skills we found in our experiment, and then explores in detail when and where these improvements occur, while also acknowledging the associated limitations. We believe that identifying these issues provides valuable context and opens avenues for future research. As this aspect is not a central focus of the paper, we have chosen to keep Figure S8 in the Supplementary Material. We also emphasize the main positive findings again in the Conclusion section to maintain the overall balance of the paper.

17. 377-379. Remove the last three sentences. They're a repetitive and weak way to finish the paper. Finish with what you showed, not what didn't work.

Thanks for the suggestion. We agree that it's already mentioned in the previous paragraph and it's better to leave them out by the end of the paper.

**Figures**
* * *
18. Figures 1 and 5: set the bottom axis at exactly zero

Thanks for reviewing the plots carefully. We did not set the y-axis to zero at the bottom because the standard deviation bars sometimes go beyond zero. So we decided to keep the axes as they are.

- 19. Figures 5 and 6: what is sea ice volume measured in? Is it actually a thickness?

  The sea ice volume in Figure 5 (Figure 6 in the updated manuscript) is the sea ice area times ice thickness over ice covered area (or total area times ice thickness per gridcell). Figure 6 (Figure 7 in the updated manuscript) shows the ice thickness per grid cell. We have it clarified in Line 140.
- 20. Figures 1 and 5: remove redundant decimal places in y axes (e.g., 6.00, 4.00, 2.00 → 6, 4, 2) Thanks for the suggestion. We have changed the label precision in y axes for Figures 1 and 5 (Figures 2 and 6 in the updated manuscript)

**21. Figure 3:**

- Why is the colormap so saturated? Increase color limit to 0.5, say?
- Redefine the acronym ACC here, since it isn't an intuitive acronym.

Thank you for the suggestions. We have changed the color bar for Figure 3 (Figure 4 in the updated manuscript; as well as Figure S6 for Arctic). We also redefined ACC in the figure captions.

**22. Figure 8:**

- Add x label ('Time (days)'?) directly to panel e, rather than just stating it in the caption. **Label for the x axis is added.**
- Make the 'DP' line for the Arctic a black dashed line. This would follow the pattern of the red and blue lines.

We chose to keep the DP line solid for both Arctic and Antarctic, as the dashed lines are meant for the reforecast experiment that's not initialized from SIC DA.

- Then consider making the legend two columns by three rows to make it really explicit how the line color/style system works.

We have changed the legend to two columns to make it clearer for the readers.

- Consider replacing 'red dot' and 'red circle' with 'filled red circle' and 'open red circle'? since the open circles aren't obvious. I couldn't initially figure out how a 'dot' was different to a 'circle'. **We've changed the caption as the reviewer suggested**
- Remove the '(red dashed lines)', '(red solid lines)' etc from the caption, which aren't necessary since these details are in the legend already.

Thanks for the suggestion. We think it's better to keep them in the caption for clarification.

23. Figure 9: Same as for Figure 8

We have edited Figure 9 (Figure 10 in the updated manuscript) as well.

24. Figure 5 error bars:

The error bars for this figure aren't appropriate. Each error bar is derived from five values. You've chosen to use ±2 standard deviations. But datasets of five are awfully small for standard deviation to be meaningful. Further, ±2 standard deviations is typically taken to be approximately equal to a 95% confidence interval (1.96 standard deviations). Of course, with only five values, you can't estimate a confidence interval that well. Could you solve all this by just using the minima and maxima of the five cases for the error bars?

Thank you for making a good point. We agree that using standard deviation values here does not serve the purpose. We have changed it to the range of values (from minimum to maximum) across the 5 years.

**25. Figure 10:**

- Different colored lines are unnecessary here since it's obvious which months are relevant from where the lines begin.

We thank the reviewer for the suggestion. We chose to keep the current color selections as they make better distinctions between each other.

- The caption is incorrect and appears to have been copy/pasted from a different figure. Thanks for catching the mistake. We have corrected the caption.
- 26. Figure S7: Same as for first comment for Figure 10.

The same edit has been done for Figure S7.

27. Typos:

330: 'from' is repeated

Captions of Figures 7 and S6: 'forcast'

Thank you for catching the typos. We have corrected them.